# The Evanescent Bouquet of Individual Bear Fingerprint

**DOI:** 10.3390/ani13020220

**Published:** 2023-01-06

**Authors:** Andrea Mazzatenta, Serena Fiorito, Roberto Guadagnini, Salvatore Genovese, Alberto Valentini, Federica Bonadiman, Sofia Guadagnini, Francesco Epifano, Adriano Mollica

**Affiliations:** 1Neurophysiology Olfactory and Chemoreceptive Laboratory, Department of Neuroscience, Imaging and Clinical Sciences, ‘G. d’Annunzio’ Chieti-Pescara University, Via dei Vestini 31, 66100 Chieti, Italy; 2Department of Pharmacy, ‘G. d’Annunzio’ Chieti-Pescara University, Via dei Vestini 31, 66100 Chieti, Italy; 3Zoolife, Via Cavalleggeri Udine 41, 38017 Mezzolombardo, Italy

**Keywords:** pheromones, European brown bear, GC/MS analysis

## Abstract

**Simple Summary:**

The evanescent chemical communication is conveyed by peculiar signals, namely pheromones. These are produced by a particular gland and are mixtures of signals that are physiologically essential to life and together constitute characteristic signatures of both species and individuals. This communication is ancient, and is little studied in mammals, mainly due to the difficulty of identifying the molecules. In this study, we collected the entire bouquet of wild bear putative pheromones from all production sites and analyzed the entire raw extract to profile the individual, species, and sex fingerprint. Our study allowed us to take a step forward, compared with the few papers in the literature, in the study of the complexity of the chemical communication of a solitary endangered species.

**Abstract:**

The evanescent and invisible communication carried by chemical signals, pheromones, or signature mixtures or, as we prefer, the pheromonal individual fingerprint, between members of the same species is poorly studied in mammals, mainly because of the lack of identification of the molecules. The difference between pheromones and the pheromonal individual fingerprint is that the former generate stereotyped innate responses while the latter requires learning, i.e., different receivers can learn different signature mixtures from the same individual. Furthermore, pheromones are usually produced by a particular gland, while the pheromonal individual fingerprint is the entire bouquet produced by the entire secreting gland of the body. In the present study, we aim to investigate the pheromonal individual fingerprint of brown bears in northern Italy. We collected the entire putative pheromone bouquet from all production sites in free-ranging bears and analyzed the entire crude extract to profile the individual fingerprint according to species-, sex- and subjective-specific characteristics. We were able to putatively characterize the brown bears’ pheromonal individual fingerprints and compare them with the partial pheromone identifications published by other studies. This work is a step forward in the study of the complexity of chemical communication, particularly in a solitary endangered species.

## 1. Introduction

The evanescent and invisible communication carried by chemical signals, pheromones, or signature mixtures between members of the same species is poorly studied in mammals, mostly because of a lack of molecules identification. The difference between pheromones and signature mixtures is that the former generate stereotyped innate responses while the latter require learning, i.e., different receivers may learn different signature mixtures from the same individual. 

However, chemical communication is among the oldest forms of non-verbal intraspecific language, which most likely evolved by sexual selection; however, it is not just about sex, and pheromones are used in a wide range of biological contexts [1]. Pheromones are typically secreted by specialized glands ranging from Harderian glands to skin and preputial glands, and they are released by fluids, such as saliva, tears, and urine [2,3,4,5,6,7]. Furthermore, pheromones may also be ‘activated’ by the activity of bacterial symbionts [8]. Consequently, a wide range of molecules are used as pheromones, covering every chemical dimension of structure, functional group, size, and combination, and they are limited only by the range of molecules organisms can produce or receive [9,10]. This range reflects the ways that molecules evolve into pheromone signals. Any molecule can potentially evolve into a pheromone [1]. 

Individual recognition by chemical stimuli, pheromones, or signature mixtures, is a fundamental phenomenon of life. Kinship recognition cues must be considered as any aspect of the phenotype and unequivocally indicate individuality and relatedness [11]. These chemical molecules are suitable for differentiated spatial and temporal communication, particularly for those species such as brown bears (Ursus arctos) that have large home ranges and a solitary, nomadic lifestyle [12]. Thus, Ursidae must rely on effective modes to communicate with conspecifics, which are still largely unexplored. Only recently, some studies have looked at signaling behaviors in bears, mostly referring to tree marking [13,14,15,16,17,18,19]. For instance, dominance and territorial signals can be left through rubbing trees in brown and black bears (Ursus americanus) [14], while sexual signals are linked to secretions from anal glands and perianal sacs [18]. Other chemo-signals may be passively deposited while the bear is walking and may originate from anal glands, urine, or a combination of the two in brown and polar bears (Ursus maritimus) [4,19]. Pheromones commonly consist of combinations of molecules that result in specific, individual signatures based on ‘signature mixtures’ [1] or, as we prefer, ‘pheromone individual fingerprints’ involved in the identification of genetic relatedness, as in giant pandas [20] and in lemurs [21], which provide fascinating examples of neurocognitive evolution. The current challenge is to unveil the chemical nature of individual pheromone fingerprints and their combination into the unique bouquets characteristic of a given individual and its species. In the present study, we investigated the distinctive individual pheromonal fingerprints of brown bears to characterize the entire bouquet of a specific subject typically released at vanishingly small quantities. This could be useful for a chemical-based monitoring of indicator species to identify biodiversity hotspots, characterize physiology, or reduce conflicts with human. 

The aim of the study is to identify the individual chemical fingerprint of the brown bear. It stands to reason that the chemo-signals characterizing a species are genetically determined and consequently stereotyped whereas sex-identifying chemo-signals, although genetically determined, may vary with sexual maturation and individual hierarchical role in line with Wyatt’s definition [1]. Accordingly, we are looking for specific compounds of individual uniqueness in accordance with the literature [1,22], which can provide an experimental demonstration of the theoretical hypothesis. Consequently, the temporally differentiated chemical communication should contain elementary information, such as species and sex, as well as advanced information about individual uniqueness that is useful to the receiving co-specific.

## 2. Materials and Methods

### 2.1. Animals and Sample Collection

We collected 32 bear (Ursus arctos arctos) samples from two males named TAZ (chip identifier number 978101083058473, aged 3 years old, and weighing 153 kg) and MAX (chip identifier number 380260043508930, aged 5 years old, and weighing 217 kg) during the capture session of the spring 2021 campaign of large-carnivore regional wildlife monitoring. The study was licensed by the Institute for Environmental Protection and Research (I.S.P.R.A.) and the Ministry of Ecological Transition, and was approved by the Wildlife Service of the Autonomous Province of Trento, prot. # 12773 date 17 March 2021. There was no need for an additional ethics committee because the samples were collected for the study by non-invasive and passive methods following authorized veterinary medical procedures and did not require extra time over the procedure time for monitoring purposes. It was carried out in compliance with the ARRIVE guidelines and all methods were in accordance with relevant guidelines and regulations [23,24]. 

The adult male bears samples were collected from different body area: conjunctiva, auricle, buccal mucosa, sternum, dorsal, penile tissue, and perineal by using swabs that were immediately immerged in Hexane GC grade (Honeywell and Merck Sigma-Aldrich, Milan, Italy). GC-MS analyses were performed on two extracts given by the union of all the samples collected from both evaluated animals.

### 2.2. Anesthesia Protocols

The anesthesia protocols were different for the two bears: Max received a solution of medetomidine (M), an alpha-2 adrenoreceptor agonist combined with tiletamine-zolazepam (TZ), and an anesthetic combination of a dissociative anesthetic and a benzodiazepine agonist; Taz received a solution of TZ with dexmedetomidine (D) and a dextrorotatory enantiomer of medetomidine. For Max, we prepared MTZ by adding 10 mg of M (Domitor® 1 mg/ml, solution for injection, Orion Pharma., Espoo, Finland) to one vial of TZ (Zoletil® 50/50 mg/ml, lyophilizate and solvent for solution for injection, Virbac SRL, Turin, Italy). We divided the solution into two 5 ml darts, each dart containing 5 mg of M and 250 mg of TZ. For Taz, we prepared DTZ by adding 5 mg of D (Dexdomitor® 0.5 mg/ml, solution for injection, Orion Pharma, Espoo, Finland) to one vial of TZ (Zoletil® 50/50 mg/ml, lyophilizate and solvent for solution for injection, Virbac SRL). We divided the solution into two 5 ml darts, each dart containing 2.5 mg of D and 250 mg of TZ. The final ratio of M to TZ was 1:50 and the ratio of D to TZ was 1:100. The combination of anesthetics was administered by remote delivery from a CO_2_-fuelled gun (Dan-Inject® Dart Guns, Austin, TX, USA). Once all procedures were completed, bears received 5 mg atipamezole (Antisedan® 5 mg/ml, Orion Corporation, Northampton, MA, USA) per mg M or 10 mg atipamezole per mg D intramuscularly to reverse the anesthesia. 

### 2.3. Extraction and Analysis

A 24 h extraction of all swabs collected from both adult male bears was performed by removing and cutting the swab cotton parts, pooling together those belonging to each animal, and transferring to 7 mL glass vials before extracting with 4 mL of pure hexane, resulting in two raw extracts. Gloves were worn when handling all swabs to avoid contamination of the samples by the collectors.

The extraction solvent evaporated to dryness and was re-suspended in 500 µL of hexane to better concentrate the sample, then vortexed for 1 min, cleaned by 0.22 µm PFTE membrane filter, and transferred into gas chromatography vial. The two crude concentrate extracts were analyzed by gas chromatography–mass spectrometry (GC-MS). 

### 2.4. GC-MS Analysis

Gas chromatography–mass spectrometry (GC-MS) analysis was carried out using a GC-MS apparatus (8860 GC with 5977B GC/MSD Agilent system, Santa Clara, CA, USA). The sample components were separated on a J&W DB-5ms Ultra Inert GC Column 30 m length × 0.25 mm i.d. capillary column coated with a 1 μm film thickness stationary phase (Agilent Technologies, Santa Clara, CA, USA). The sample volume of 0.2 μL was injected using AOC-20i + s auto injector. The injection port was set at 250 °C in pulsed splitless mode. The GC oven temperature was programmed as follows: 2 min at 50 °C, increased by 5 °C per minute at 300 °C (with 10 min hold at 300 °C). Solvent delay was set to 3 min. 

The ion source temperature in the MS was set at 230 °C, while the interface was set at 280 °C. Total ion chromatogram (TIC) was created for m/z range 40–450. GC peaks were identified by comparing their mass spectra to the database of the National Institute of Standards and Technology (NIST 11, Mass Spectral Library 2011/EPA/NIH). The data were analyzed using commercial software OriginPro 2018 (OriginLab Corporation, Northampton, MA, USA) and Jamovi (Version 1.6, www.jamovi.org) (Accessed on 18 June 2021) [25].

## 3. Results

### 3.1. Brown Bear Individual Compounds Isolation

We identified a total of 32 compounds, comprising large molecular weight alcohols, acids, steroids, and esters, which are listed in Table 1, ranging from 254 to 650 in molecular mass, with a mean of 371.64 ± 90.79 SD, in the swabs of sampled free-ranging male bears (the number of compounds identified were 19 and 22 for ‘Tax’ and ‘Max’, respectively) present in the whole samples, irrespective of the body district from which were swabbed, Figure 1 shows the GC/MS chromatograms. Pareto analysis points out that 90% of the compounds range from 354 to 454. 

Furthermore, we have been able to discriminate two different putative chemo-signals fingerprints from different body areas of the sampled bears, which shared nine compounds that are listed in Table 1.

### 3.2. Brown Bear Putative Species vs. Individual Chemo-Signals

Comparison of the chemical fingerprints of the brown bears’ bodies revealed 10 stereotypical compounds characteristic of the species and sexual identity, whereas the 16 compounds of individual uniqueness were variable between individuals (Figure 2A). Individual differences in the distribution, density, and abundance of isolated compounds are highlighted in Figure 2B. The normality test is *p* = 0.163, W = 0.956. Consequently, the non-normal distributions were compared with the non-parametric Kruskal–Wallis analysis, which returns *p* = 0.384, X2 = 0.759, d.f. =1. The most differences were found at RT 37.5–40, 42.5–47.5, and 52.5–55. The polynomial fit of the ‘Max’ distribution was R2 = 0.42, while ‘Taz’ was R2 = 0.54.

### 3.3. Brown Bear Population and Body District Putative Pheromones Comparison

We also compared the total compounds extracted in the present study with those found in the literature [4]. Both studies used similar techniques to isolate putative brown bear chemo-signals. The discrepancies were in the geographical/genetic differences of the populations and in the body district sampled. The Polish study only isolated compounds from the pedal region, whereas in the current study, we sampled from the putative chemo-signals emission sites of the whole body, with the exception of the pedal glands, which is not clearly visible in the field and would result in a generic ‘mud’ sampling, potentially contaminated with aromatic compounds from plants and soil. Figure 3A shows comparisons between Polish and Italian brown bear populations and their body districts. The chemical fingerprints of the brown bear and body district populations revealed a bimodal distribution in both groups: ‘Polish’ R2 = 0.55 and ‘Italian’ is R2 = 0.50, as shown in Figure 3B. The normality test is *p* = 0.115, W = 0.968. Non-parametric Kruskal–Wallis analysis returns *p* < 0.001, X2 = 42.1, and df = 1, with the most noticeable difference being that the distributions appear to be inverted.

## 4. Discussion

To the best of our knowledge, two works concern putative bear pheromones isolation: The first showed that brown bear anal gland secretion contains a high number of compounds with high molecular mass above 300 g/mol, indicating its possible use in long-lasting scent marks [26]; the second identified a series of compounds in the pedal region ranging from 116 to 468 g/mol molecular mass, with the majority below 300 g/mol, and with analogous physiological and behavioral effects [4]. 

Mammalian ‘pheromonal’ or ‘signature mixture’ secretions typically contain a wide range of compounds, including aldehydes, alcohols, ketones, esters, sterols, and acids [1]. The difference between pheromones and signature mixtures is that the former generate stereotyped innate responses while the latter theoretically require learning [1]. This concept is highly supportable, regardless of the terminology used to identify it, e.g., ‘signature mixture’ or, as we prefer, ‘pheromonal individual fingerprint’, because it is the result of total body secretion and individual recognition is of utmost importance for territoriality and reproduction, e.g., both to attract an unrelated partner of the opposite sex and to avoid a certain stronger subject of the same sex. Accordingly, in our study we investigated the entire pheromonal individual fingerprint of brown bears, the molecular weight of which was found to be in the range from 354 to 454 g/mol and which comprised high-molecular-weight alcohols, acids, steroids, and esters. Furthermore, this result is in line with previous studies on Panda pheromone [27,28] and the chemical scent of brown bears; however, these studies were biased because they focused exclusively on the perianal or pedal glands, whereas we studied the whole chemical fingerprint [4,26]. 

The volatility of mammalian pheromone usually decreases with increasing molecular mass, although some compounds with relatively high molecular masses may be sufficiently volatile under certain environmental conditions as well due to the degrading action of bacteria [29]. With increasing size or polarity, the rate of evaporation decreases, and the signal can be emitted for a longer period [30]. The more volatile compounds identified in this study belong to groups of fatty acids, such as palmitic (R.T. 38.202) and stearic (R.T. 42.088) acids, which were found to be shared and abundant in both analyzed samples. Palmitic and stearic acids, respectively, are the precursors of palmitoleic acid (R.T. 37.793), which is found in the Max bear, and oleic acid (R.T. 41.736), which is present in both Taz and Max and represents the most abundant monoenoic fatty acid in plant and animal tissues, both in structural lipids, sebum secretion, and in depot fats. Furthermore, vaccenic acid is a natural monoenoic trans fatty acid, usually present as a minor component of most plant and animal tissues, which has been recently recognized as one of those responsible for the human body odor developed with aging [31]. 

Concerning the alkane family, their presence may be attributed in prevalence to dietary intake; however, some substances, such as n-tetracosane and pentacosane, have been identified as constituent of human body scent, and thus it is reasonable to consider these molecules as resulting from catabolism and important constituents of the bear pheromone bouquet [32]. Three major fatty acids were identified: cis-11-octadecenoic acid (R.T. 41.617) (also known as cis-vaccenic acid), palmitic acid (R.T. 38.202), and cis-7-hexadecenoic acid (R.T. 37.793). Palmitic acid concentration was highest in the summer bears. In the European brown bear (Ursus arctos arctos), palmitic acid in plasma was increased in the winter, while stearic acid was decreased in the winter [33]. Among the unsaturated fatty acids, 13(Z)-docosenoic acid is a 22-carbon monounsaturated fatty acid found predominantly in canola oil, which is metabolized to oleic acid. 7-methyl-Z-tetradecen-1-ol-acetate (C17) (Taz R.T. 39.207 and Max R.T. 39.211) is a bioactive phytochemical compound identified in the methanolic extract of Mentha viridis [34], and it was also characterized as lepidoptera pheromones [35]. 1-heptatriacotanol (R.T. 44.993) is a phytochemical constituent of *Blepharis maderaspatensis*, *Artemisia annua*, and *Achillea filipendulina* (L.) leaves [36]. 3-(octadecyloxy) propyl ester oleic acid (C39) (R.T. 48.507) is a bioactive phytochemical compound [37]. Ethylisoallocholate (R.T. 49.495) and 2,3-dihydroxypropyl ester octadecanoid acid (1α-monostearin) (R. T. 51.388) are also phytochemical constituents [38,39,40]. Regarding the last part of the chromatograms, the main peak (R.T. 59.770 min for Taz and 59.836 min for Max) is represented by cholesterol. Several other steroidal substances have been also identified, such as cholesterol esters, and their unsaturated isomers. Cholesterol and its esterified precursors play an important role in the biosynthesis of steroid hormones: progestogens, glucocorticoids, mineralocorticoids, androgens, and estrogens. Steroidogenic tissues are involved in multiple pathways to assure the constant supply of cholesterol needed to maintain optimum steroid synthesis. The identification of cholesterol and its esterified counterparts content in bears could represent another important aspect for the detection of seasonal changes [41]. Specific molecules, such as 3(β)-cholesta-4,6-dien-3-ol (R.T. 54.332) and cholesta-3,5-diene C37 (R. T. 54.736), have been previously identified as individual constituents of Rhesus macaques [42]. 

As reported by Shapiro B. [43], mammals are capable of synthetizing de novo saturated and monosaturated fatty acids, which, as well as diet composition, has a strong influence on the whole fatty acids pool and related esters found in the sampled bears. Moreover, the plasma fatty acids variations change in relation to the different periods of the year, especially during the denning and winter sleep periods [33].

## 5. Conclusions

In this work we investigated the individual uniqueness of the brown bear’s chemical phenotype as a reliable species, sexual, and individual marker. Distinguishing between pheromones and signature mixtures, or individual pheromone fingerprints as we prefer, is an upcoming new scientific question stimulated by Wyatt [1] that goes beyond the original definition of pheromone uniqueness [44]. One way to investigate this would be to compare species that live solitarily to see if individuals emit a bouquet of substances that can be traced back to species, sex, and individuality. 

In conclusion, our examination of swabs from specimens of free-ranging male bears found something related to fatty acid composition, dietary intake of elements, and steroid hormone metabolism, yielding a signature that can simply be referred to mixtures that are sufficiently stable and individually different to allow the same individual to be recognized on another occasion by individual pheromonal fingerprinting.

## Figures and Tables

**Figure 1 animals-13-00220-f001:**
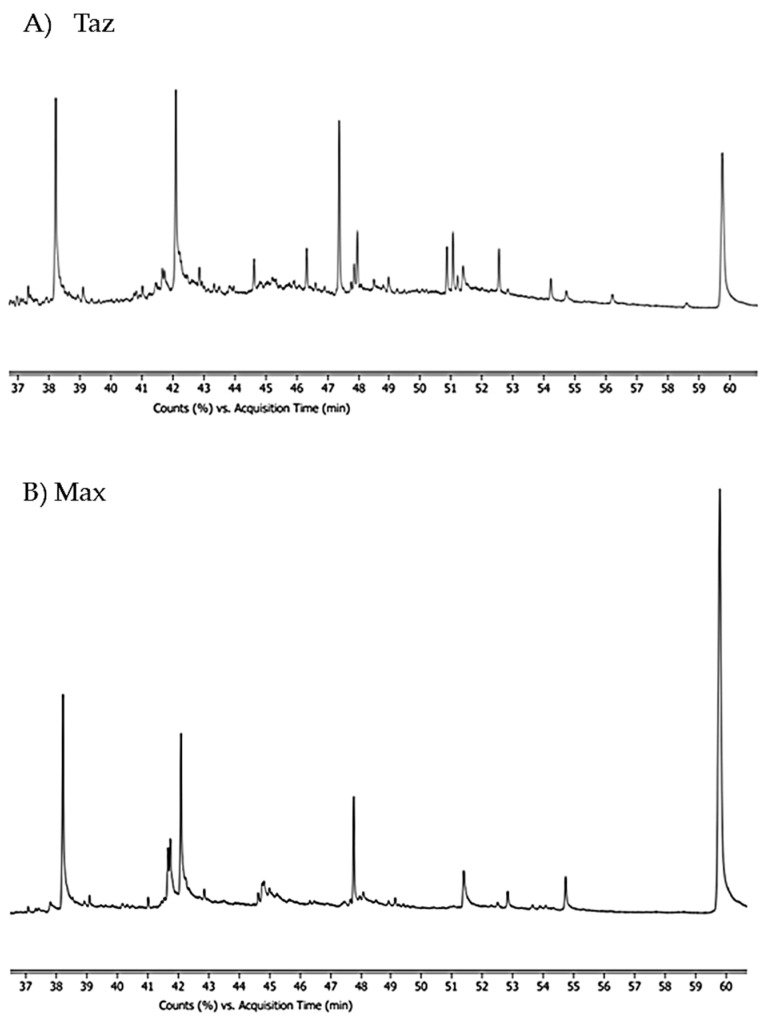
Comparison of GC/MS chromatograms of the two sampled adult male brown bears (**A**) TAZ and (**B**) MAX extracted with hexane. The x-axis is the retention time in minutes.

**Figure 2 animals-13-00220-f002:**
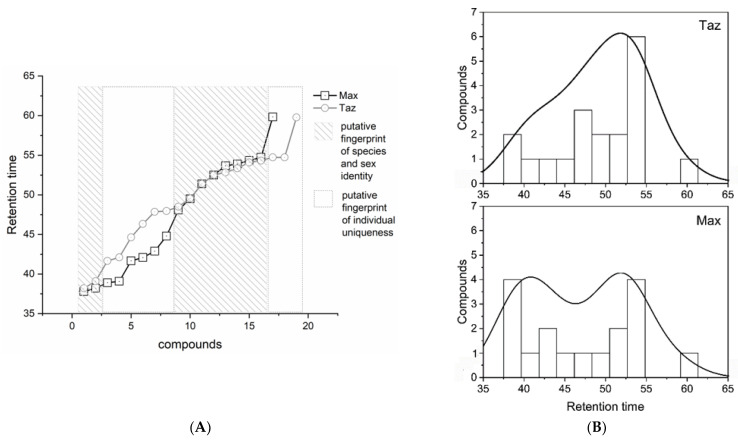
(**A**) Chemical fingerprints of the brown bear’s body reveal species and sex identity with respect to individual uniqueness; (**B**) comparison of the distribution, density, and abundance of individual compounds.

**Figure 3 animals-13-00220-f003:**
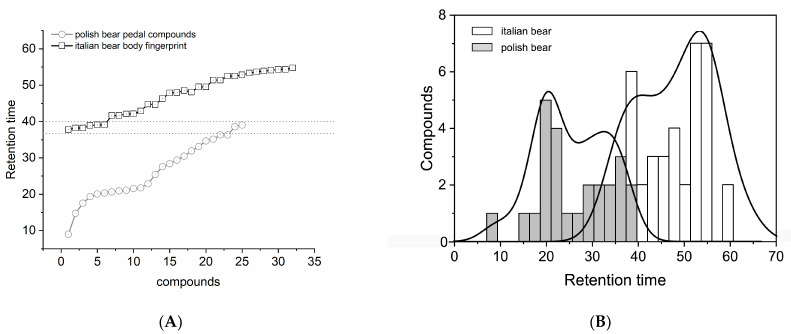
(**A**) Comparison of brown bear’s compounds extracted from polish population by pedal glands and Italian population by total body district; (**B**) distribution, density, and abundance comparison.

**Table 1 animals-13-00220-t001:** List of individual compounds identified in all samples collected from the different body areas of TAZ and MAX brown bears.

R.T. (min)	Identified Compounds (TAZ Bear)	R.T. (min)	Identified Compounds (MAX Bear)	M.W.	Matching Percentage (NIST Library)	Chemical Class
		37.793	9(Z)-hexadecenoic acid (palmitoleic acid) (C16:1)	254	20	Carboxylic acid
38.202	Palmitic acid (n-hexadecenoic acid) (C16:0)		Palmitic acid (n-hexadecenoic acid) (C16:0)	256	70.3	Carboxylic acid
		38.907	Ethylpalmitate (C18)	284	40	Carboxylic acid ester
39.207	7-methyl-Z-tetradecen-1-ol-acetate (C17)	39.211	7-methyl-Z-tetradecen-1-ol-acetate (C17)	268.43	13	Acetate ester
		41.617	cis-vaccenic acid (C18:1) (cis-11-octadecenoic acid)	282	15	Carboxylic acid
41.655	Oleic acid (C18)	41.736	Oleic acid (18:1n-9) (cis-9-octadecenoic acid)	282	20	Carboxylic acid
42.093	Octadecanoic acid (stearic acid) (C18:0)	42.088	Octadecanoic acid (stearic acid) (C18:0)	284	70.6	Carboxylic acid
44.626	Heptacosane (C27)	44.626	Heptacosane (C27)	380.7	12.5	Alkane
44.689	(Z)-13-docosenoic acid (C22)	44.793	(Z)-13-docosenoic acid (C22)	338	35	Carboxylic acid
46.326	Tetracosane (C24)			338	41.3	Alkane
47.964	Pentacosane (C25)			352	13	Alkane
48.507	3-(octadecyloxy) propyl ester oleic acid (C39)			592	38.8	Carboxylic acid ester
		49.495	Ethylisoallocholate (C26)	436	46.9	Steroid
		51.384	2-hydroxy-1-(hydroxymethyl) ethyl ester (octadecanoic acid) (C21)	358	40	Carboxylic acid ester
51.388	2,3-dihydroxypropyl ester octadecanoid acid (1α-monostearin) (C21)			358	45.2	Carboxylic acid ester
52.508	Cholesterol (C27)			386	49.2	Steroid
52.826	Ethyl iso allocholate (C26)			436	35	Steroid ester
53.346	5α, 14 β cholest 9(11)-ene (C27)			370	27.8	Steroid
		53.650	Cholest-1-ene (C27)	370	57.6	Steroid
		53.889	Cholest-3-ene (C27)	370	30	Steroid
		53.893	Cholest-5-ene (C27)	370	30	Steroid
54.089	Cholest-5-ene-3-ol (3β), 9-octadecenoate (Z) (C45)			650	11.4	Steroid ester
54.332	Cholesta-4,6-dien-3-ol (3β) (C27)	54.331	Cholesta-4,6-dien-3-ol (3β) (C27)	384	40	Steroid
		54.689	Cholesteryl formate (C34)	490	14	Steroid ester
54.736	Cholesta-3,5-diene (C27)	54.735	Cholesta-3,5-diene (C27)	368	35	Steroid
		54.736	Cholesteryl benzoate (C34)	490	20	Steroid ester
54.741	5-cholesten-(3β)-yl-isobutyl carbonate (C32)			486	14.9	Steroid ester
59.770	Psi-cholesterol (C27)	59.836	Psi-cholesterol (C27)	386	36.2	Steroid

## Data Availability

Data available at University Veterinary Service repository.

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
