# Peer review of "The Evanescent Bouquet of Individual Bear Fingerprint"

_animals, 2023, doi:10.3390/ani13020220_

Round 1

Reviewer 1 Report

Dear Authors,

Please improve the marked sentences (file).

1. What is the main question addressed by the research?
The research is described in the aim and in table 1.

2. Do you consider the topic original or relevant in the field? Does it address a specific gap in the field?
The gap is the chemo signals of animals in the wildlife because it is important to identify animals with animal welfare. Therefore this research is important.

3. What does it add to the subject area compared with other published material?
The different compounds listed in table 1 is different to others.

4. What specific improvements should the authors consider regarding the methodology? What further controls should be considered?
It is described.

5. Are the conclusions consistent with the evidence and arguments presented and do they address the main question posed?
Correct.

6. Are the references appropriate?
Correct.

7. Please include any additional comments on the tables and figures.
The figures and the table are correct.

Author Response

Reviewer 1

Dear Authors,

Please improve the marked sentences (file).

According the Referee we have edited all.

  1. What is the main question addressed by the research?
    The research is described in the aim and in table 1.

    2. Do you consider the topic original or relevant in the field? Does it address a specific gap in the field?
    The gap is the chemo signals of animals in the wildlife because it is important to identify animals with animal welfare. Therefore this research is important.

    3. What does it add to the subject area compared with other published material?
    The different compounds listed in table 1 is different to others.

    4. What specific improvements should the authors consider regarding the methodology? What further controls should be considered?
    It is described.

    5. Are the conclusions consistent with the evidence and arguments presented and do they address the main question posed?
    Correct.

    6. Are the references appropriate?
    Correct.

    7. Please include any additional comments on the tables and figures.
    The figures and the table are correct.

We would like to thanks the Reviewer.

Reviewer 2 Report

“The Evanescent Bouquet of the Bear Individual Fingerprint” is an article that had the purpose of identifying, by gas chromatography-mass spectrometry analysis, the substances (pheromones) collected in various body sites of two male brown bears, in order to characterize a typical bouquet specific to the animal considered. Furthermore, a comparison between the results obtained by the authors and those previously published by other authors was performed.

The work is very interesting and provides a new element in understanding the relationships between the composition of the unique bouquet typical of the animal and the social role and sexual behavior of the animal.

However, there are some issues that need to be considered and clarified.

I would recommend remodulating the final part of the introduction: it should set out the aims of the work and possibly the methods used, but summarizing and commenting on some results should be avoided.give new information about animal chemical communication. Furthermore, a comparison between the results obtained by the authors and those previously published by other authors

and give two brown bears the pheromones (chemical compounds) that make up the individual fingerprint of two brown bears and to compare the results obtained with those found in the bibliography.

I would recommend remodulating the final part of the introduction: it should set out the aims of the work and possibly the methods used but summarizing and commenting on some results should be avoided.

Please, for Materials and Methods and Results paragraphs, use the past tense for the verbs.

I would suggest introducing subparagraphs in Mat and Met, for instance for “Extraction and analysis”, “GC-MS analysis” and add new ones, as “Animals and sample collection” and so on.

About sampling methods: were they performed with the use of gloves to avoid contamination of the samples by the collectors?

The treatment of the samples taken is not clear to me: were they evaluated individually by GC-MS or was a single extract given by the union of all the samples taken from each animal evaluated? If the latter is true, did you not feel that making individual levy determinations would yield more interesting results?

Bear sampling in Poland was limited to pedal sampling: if you were going to compare the results of your study with those of the Polish bear, perhaps it would have been appropriate to take pedal samples in your studio as well. It would have been more appropriate.

Finally, please, check carefully the references: the first two references have the same number.

Author Response

Reviewer 2

“The Evanescent Bouquet of the Bear Individual Fingerprint” is an article that had the purpose of identifying, by gas chromatography-mass spectrometry analysis, the substances (pheromones) collected in various body sites of two male brown bears, in order to characterize a typical bouquet specific to the animal considered. Furthermore, a comparison between the results obtained by the authors and those previously published by other authors was performed.

The work is very interesting and provides a new element in understanding the relationships between the composition of the unique bouquet typical of the animal and the social role and sexual behavior of the animal.

However, there are some issues that need to be considered and clarified.

  • I would recommend remodulating the final part of the introduction: it should set out the aims of the work and possibly the methods used but summarizing and commenting on some results should be avoided give new information about animal chemical communication. Furthermore, a comparison between the results obtained by the authors and those previously published by other authors and give two brown bears the pheromones (chemical compounds) that make up the individual fingerprint of two brown bears and to compare the results obtained with those found in the bibliography.

According the Referee suggestion we remodulate the the final part of the introduction see line 79-89.

  • Please, for Materials and Methods and Results paragraphs, use the past tense for the verbs.

According the Referee we have correct it.

  • I would suggest introducing subparagraphs in Mat and Met, for instance for “Extraction and analysis”, “GC-MS analysis” and add new ones, as “Animals and sample collection” and so on.

According the Referee we have edited.

  • About sampling methods: were they performed with the use of gloves to avoid contamination of the samples by the collectors?

According the Referee we have added to the text see Line 103-107

  • The treatment of the samples taken is not clear to me: were they evaluated individually by GC-MS or was a single extract given by the union of all the samples taken from each animal evaluated? If the latter is true, did you not feel that making individual levy determinations would yield more interesting results?

According the Referee we explain that the GC-MS analysis were performed on two extracts given by the union of all the samples collected from both evaluated animals. Now has been included in the text. To explain the second question we decided to concentrate the extract for a more suitable evaluation since the aim of this study concerns the entire bouquet of wild bear pheromones (line 127-129).

  • Bear sampling in Poland was limited to pedal sampling: if you were going to compare the results of your study with those of the Polish bear, perhaps it would have been appropriate to take pedal samples in your studio as well. It would have been more appropriate.

The referee is undoubtedly right, but the duct of the pedal glands is not clearly visible in the field experience so a generic ‘mud’ sampling would be obtained. Thus, the sample is potentially contaminated with aromatic compounds from plants and soil. Instead, we have collected a number of body glands with a reduced possibility of environmental contamination. Please note we introduce a sentence to explai n this see line 193-196.

  • Finally, please, carefully check the references: the first two references have the same number.

done

Round 2

Reviewer 2 Report

Dear Authors,

Thank you for your reply.

I am satisfied with your corrections and your reasons, which emerged from my observations, have been explained in a convincing way.

In my opinion, therefore, the work can be published in Animals in the present form.